# Humans underestimate the movement range of their own hands
Artur Pilacinski [1] ✉, Antoine Vandenberghe[2,3], Gabriella Andrietta[4] & Gilles Vannuscorps [2,3] ✉

Motor planning and motor imagery are assumed to use veridical internal representations of the biomechanical properties of our limbs. Here, we report that people underestimate their hands' range of motion. We used two tasks probing representations of own motion range, estimation and imagery, in which participants were supposed to judge their rotational hand movement ranges. In both tasks participants' judgments were underestimated in three out of four cardinal directions. We suggest that this representational bias provides an optimal balance between movement efficiency and safety in face of the inherently stochastic nature of movement execution.

Internal representations of our limbs' range of motion are pivotal to our ability to interact efficiently and safely with our environment. Imagine filling a glass with water. To efficiently plan and execute this action, our brain needs to factor in the limitation of our wrist's rotation to decide whether, and if so when, maintaining a steady stream will require engaging an additional rotation of the shoulder. Given that adults have had years of exploration and practice to refine and update their knowledge about how their hands move, one could assume that we all know our limbs' range of motion with unparalleled accuracy.

In line with this idea, it is often assumed that motor planning and imagery use veridical representations of one's own biomechanical constraints, acquired through motor experience and proprioceptive feedback[1–4]. For example, during movement planning, biomechanical cost of a movement is automatically accounted for, suggesting that internal models (e.g. body schema) must possess veridical biomechanical representations of the body when planning a movement[3,4]. Motor imagery is a process of consciously simulating movements, albeit without overtly preparing them and, according to major theories, motor imagery is based on real movement repertoire, reflecting its properties such as anatomy and biomechanical constraints of body movements[1,2]. Importantly, both motor planning and imagery share neural substrates[1] and engage conscious awareness, most likely to enable efficient action monitoring[5,6].

Yet, whether we possess a veridical representation of our own range of motion remains unclear. Indeed, several recent studies have shown that humans have surprisingly distorted representations of their hands' position, size[7–9] and weight[10]. Therefore, the goal of this study was to explore the quality of hand movement range representations. How accurate are they? And, more importantly, are they consistently biased?

Distortions and biases have a long history of providing a window into the working of the mind/brain. For instance, visual illusions have significantly contributed to reveal the assumptions and heuristics that the brain

uses to interpret sensory information. Likewise, biases in body representation have for long provided important insights into action control and planning. For instance, the systematic underestimation of our hands' weight, recently reported by Ferre et al. has been suggested to play an adaptive role in balancing motivation for action with the need to rest[10]. Hence, by exploring potential biases in movement range representation, we hoped to contribute new insight about the constraints within which action planning operates and the trade-offs that underlie its efficiency.

Across two experiments, we explored participants' representation of their hands' range of motion, focusing on simple wrist rotations rather than on complex, multi-joint actions[11]. In the first experiment, we used a judgment task. Sixty participants were asked to estimate and then execute the maximal amplitude of rotational movement that they could perform in the four cardinal directions with their left and right wrists (abduction/adduction/flexion/extension; see Fig. 1 A). In a second experiment, we sought to corroborate the findings from the first experiment in a slightly different, motor imagery task. Twenty-five participants were asked to imagine the maximal amplitude of wrist abduction and adduction that they could perform. This allowed exploring the properties of hand movement range representations used when participants are engaged in explicit motor imagery, which is often assumed to reflect motor plans without overt movements[1,2].

## Methods
### Participants
Sixty volunteers participated in Experiment 1. All participants had normal or corrected-to-normal vision, none reported a history of wrist injury, and only one reported performing frequent flexibility exercises. That participant was excluded from the analyses. The final sample was therefore composed of 59 participants (*Mean age* = 21.3, *SD* = 2.5, 55 right-handed, 48 men and 11 women).

[1]Medical Faculty, Ruhr University Bochum, Bochum, Germany. [2]Psychological Sciences Research Institute, Université catholique de Louvain, Louvain-la-Neuve, Louvain, Belgium. [3]Institute of Neuroscience, Université catholique de Louvain, Louvain-la-Neuve, Louvain, Belgium. [4]CINEICC, University of Coimbra, Coimbra, Portugal. ✉e-mail: art.pilacinski@gmail.com; gilles.vannuscorps@uclouvain.be

**Fig. 1 | Task depiction and main results.**
**A** Illustration of the four rotational hand movements. **B** Group averages show underestimation of actual range of motion in Experiment 1 (real – estimated range, in degrees; *n* = 59). **C** Group averages show underestimation of the range of motion in motor imagery (Experiment 2; *n* = 25). Horizontal lines are medians; X denotes average; whiskers denote interquartile range, dots denote outliers. One sample t-test, 2-tailed; *$p < 0.05$, **$p < 0.01$, ***$p < 0.001$.

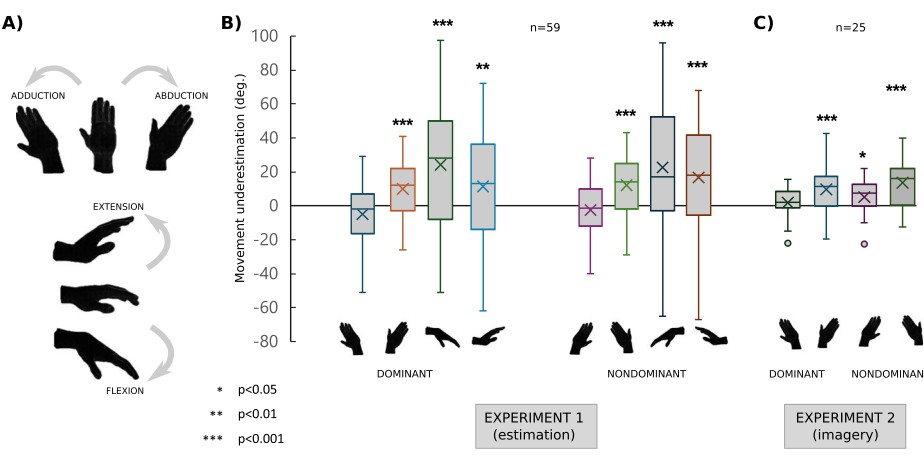

Twenty-five volunteers took part in Experiment 2 (age range 21–40, 22 right-handed, 8 men, 17 women). The participants were informed only about the general goal of the experiment (judgments about human hand motion) and its approximate structure and duration. No participants reported past wrist injuries. We additionally asked subjects whether they perform any manual activities that would influence their estimations. None of the subjects reported such a thing.

Participants gave written informed consent and received course credit for their participation, in accordance with the procedures of the study approved by the biomedical ethics committee of the Cliniques Universitaires Saint-Luc, Brussels, Belgium. Sampling for both studies was done per convenience, with basic pre-screening (see above). We used sample sizes similar or higher to those used before for similar studies on estimation of body properties[8–10]. The study was not preregistered.

## Data analysis
All data was analyzed and visualized with *Jamovi* (The *jamovi* project (2024). *jamovi* (Version 2.3.28). Retrieved from https://www.jamovi.org) and figures were created using *Inkscape* (*Inkscape Project. (2024). Inkscape. Retrieved from* https://inkscape.org).

## Materials and procedure
**Experiment 1**. The experiment comprised two tasks in which participants had to first estimate the amplitude of imagined wrist movements, and then to perform them in the four cardinal directions (abduction/adduction/flexion/extension) with a maximal amplitude. As we expected that estimations might differ for dominant and non-dominant hands due to their different involvement in precise actions, we conducted measurements for both hands separately.

The setup for both tasks was the same. Participants sat on a chair in front of a table and were asked to place one arm and hand in a precise "resting position" on the table. The arm had to be oriented in the sagittal plane, aligned with the axis of reference of a 360 degrees protractor drawn on the table. Accurate positioning was ensured by making sure that a double-sided adhesive pad placed on the participant's olecranon overlapped perfectly with another one positioned on the axis of reference of the protractor. For adduction and abduction movements, the hand in pronation was placed such that a double-sided adhesive pad placed on the surface of the transverse carpal ligament between the thenar and hypothenar eminences overlapped precisely with another one placed on the centre of the protractor. For flexion and extension movements, the hand in neutral orientation was placed such that a double-sided adhesive pad placed on the ulnar styloid overlapped precisely with another one placed on the centre of the protractor. In addition, participants' hand was attached to a hand-sized rigid plate to prevent additional movements of the fingers (See Supplementary Fig. S1). For all movements, the fingers remained packed together.

Once in "resting position", participants started the estimation task. There were 8 conditions: 2 hands (left and right), 2 movement types (adduction/abduction and flexion/extension) and 2 movements for each movement type (e.g. adduction vs abduction). The order of hands and movement types were counterbalanced across participants.

For each of these 8 conditions, the experimenter first started by showing a short (2–4 s) video-clip illustrating the relevant movement of the hand to the participant. The movements illustrated in the videos were of only a few degrees and the participants were informed that these videos did not illustrate a natural amplitude of movement, that the objective was only to make sure that the participant understood the movement that we were interested in. Once the movement instruction was clear, the experimenter asked participants to decide whether they would be able to reach a dot drawn on the protractor with their middle finger by rotating their own wrist in that direction (the instructed answer was only "yes/no"). For half of the participants, the experimenter started with a dot corresponding to an extremely large rotation angle (90° of amplitude for adduction and abduction, and 120° for flexion and extension) and then proceeded in a descending order by steps of 3 degrees. The first "yes" response of these participants was recorded as their estimated maximal wrist rotation. For the other half of the participants, the experimenter started with a dot that corresponded to 3 degrees of rotation and then proceeded in an ascending order by steps of 3 degrees. The last "yes" response of these participants was recorded as their estimated maximal wrist rotation. Later on, participants were asked to perform real movements. They performed the eight movements in the same order as in the estimation task. Participants were asked to perform the largest wrist rotation possible and to maintain that position as the experimenter was recording the last reached dot. As goniometry measurements are known to be variable[12] the movement was repeated three times consecutively. The median of the three measures was computed and recorded as the index of maximal amplitude.

**Experiment 2**. The global wrist motion is usually considered as the motion of the second and/or third metacarpal with respect to the radius and the hand motion is commonly considered as represented by the second or third metacarpals[13]. A palpation method was used to center the participant's hand on the scale by finding crucifixion fossa, a soft depression posterior to the third metacarpal, anterior dorsal end of the radius and medial to the lunate[14]. This depression and the tip of the middle finger were used as references to center the palm in a prone position on the scale.

The participant's hand was placed on a radial scale comprising a protractor with no numeric labels positioned on a desk. The participants were asked to imagine lateral hand rotations, abduction, and adduction. The experimenter, sitting in front of the participant, indicated left and right directions with their finger in order to instruct the movement direction

without performing their own hand rotation. We decided for this to avoid providing any hints regarding hand motion range (cf. Experiment 1). The participants were asked to imagine how far they could perform those movements and to point with their non-tested hand, on the scale, where their middle finger would land after maximal rotation. Each estimation was repeated twice, interleaved with other estimations, in a counterbalanced fashion. As there were no substantial differences between estimations in these repetitions (on average less than 3 deg.), we used a mean value for analyses. After the imagery task, the participants performed the relevant movements. The positions of the middle finger were used to find the participant's real movement angle.

### Reporting summary

Further information on research design is available in the Nature Portfolio Reporting Summary linked to this article.

## Results

### Experiment 1

The first glance at the data revealed clear discrepancies between the absolute values of the actual and imagined wrist rotation amplitude as uncovered by one-sample, two-tailed t-test performed independently for each movement: all $t_{(1, 58)} > 9.51$, all $p < 0.001$, all $d > 1.24$ (see Table 1 for full statistics). We assessed normality using K-S test which didn't show violation to the normality assumption. We tested whether these deviations varied across different types of movement. We then ran an ANOVA including the factors "hand" (dominant/non-dominant) and "type of movement" (adduction/abduction/flexion/extension) as a within-subject factors. The analysis revealed an effect of the type of movement, $F_{(1, 174)} = 27.21$, $p < 0.001$, $eta^2_G = 0.113$, further assessed through post-hoc two-tailed t-test as significant differences between adduction/abduction vs. flexion/extension, showing that flexion/extension yield significantly higher inaccuracies than adduction/abduction, apart from abduction vs. extension (see Table 2 for details). The average sizes of those inaccuracies in the dominant hand were the following: flexion: 35 deg., extension: 28.5 deg., abduction: 17 deg., adduction: 15 deg. See Supplementary Table S1 for descriptive statistics.

In the next step, we aimed at characterizing the direction of discrepancies (over- vs. underestimation) between the true and the imagined maximal amplitudes of movements. To do so, we first compared the true

maximal amplitude of movements to the imagined amplitude of movements using paired t-tests. We found that estimated rotations are significantly smaller than real ones across all types of movement (all $t_{(1, 58)} > 2.77$, $p < 0.008$, all $d > 0.36$) apart of adduction (dominant hand: $t_{(1, 58)} = -1.97$, $p = 0.054$, $d = -0.256$; non-dominant hand: $t_{(1, 58)} = -1.08$, $p = 0.29$, $d = -0.14$). See Table 3 for detailed statistics. These underestimations were higher in the non-dominant hand ($t_{(1, 58)} = -2.87$, $p = 0.006$).

### Experiment 2

To test the accuracy of imagined movement estimations we calculated absolute values of difference between estimated and real wrist rotation values and performed a one sample two-tailed t-test. We first assessed normality using K-S test which didn't show violation to the normality assumption. We found that subjects are significantly inaccurate when estimating their wrist movement range in all conditions (Table 4) as measured by one-sample t-test (all $t_{(1, 24)} > 5.93$; all $p < 0.001$, all $d > 1.19$).

Next, we asked if there is a systematic bias in these estimates of movement with respect to movement direction. For this, we compared real and imagined rotation values with a paired t-test (two-tailed). We found that subjects on average underestimate the abduction movement in both right and left hand (all $t_{(1, 24)} > 3.85$, all $p < 0.001$, all $d > 0.77$). Adduction movement was significantly underestimated in the non-dominant hand ($t_{(1, 24)} = 2.32$, $p = 0.029$, $d = 0.46$), but there was no statistically significant effect in the dominant hand ($t_{(1, 24)} = 0.96$, $p = 0.35$, $d = 0.191$). Similar to Experiment 1, the underestimations were larger for non-dominant hand ($t_{(1, 24)} = -3,58$, $p = 0.022$). See Table 5 for detailed statistics.

## Discussion

It was previously reported that representations of several static properties of our hands, such as their size and weight, are systematically biased[7,8,10]. Here, we report that the representation of our hands' range of movement is also biased. We conducted two experiments exploring participants' representation of their two hands' range of rotational movement through judgments (Experiment 1) and motor imagery (Experiment 2) and found an underestimation in three of the four tested cardinal directions (abduction/flexion/extension).

Why would we underestimate our limbs' range of motion? Motor planning operates under uncertainty, dealing with factors such as neural noise, muscle fatigue, stress, muscle fibre composition, and motor unit recruitment patterns – often collectively referred to as "sensorimotor noise"[15,16]. This inescapable unpredictability and variability of movement forces our motor system to consider the impact of putative errors when planning actions. Of course, planning an undershot movement might unnecessarily increase the metabolic and cognitive cost of movement when the target/goal lies within the actual range of motion. In such case, underestimation leads to recruiting more joints than required or to adding a correcting movement. This additional cost, however, appears very small compared to the potential cost of errors if the action plan was originally based on a representation close to (or exceeding) one's real maximal range of motion. In this case, an error could not only lead to a complete failure of the action (i.e., to halting of the movement induced by feedback from stretch receptors), but also strain on the bones, muscles, tendons, and ligaments

**Table 1 | Differences between the absolute values of the actual and estimated wrist rotation amplitude in Experiment 1. One sample T-test**

| Movement | t | df | p | d | Lower 95% CI | Upper 95% CI |
|---|---|---|---|---|---|---|
| Adduction | 10.4 | 59 | <0.001 | 1.34 | 0.988 | 1.69 |
| Abduction | 13 | 59 | <0.001 | 1.68 | 1.284 | 2.07 |
| Flexion | 10.4 | 59 | <0.001 | 1.35 | 0.995 | 1.7 |
| Extension | 12.4 | 59 | <0.001 | 1.6 | 1.21 | 1.98 |

Note. $H_a$ $\mu \neq 0$.

**Table 2 | Differences between adduction/abduction vs. flexion/extension. Paired samples T-Test**

| Movement 1 | Movement 2 | t | df | p | Mean diff. | d | Lower 95% CI | Upper 95% CI |
|---|---|---|---|---|---|---|---|---|
| Adduction | Abduction | −9.36 | 59 | < .001 | −14.55 | −1.209 | −1.539 | −0.872 |
| | Flexion | −7.55 | 59 | < .001 | −26.93 | −0.975 | −1.28 | −0.664 |
| | Extension | −6.56 | 59 | < .001 | −17.65 | −0.847 | −1.14 | −0.549 |
| Abduction | Flexion | −3.51 | 59 | < .001 | −12.38 | −0.454 | −0.718 | −0.186 |
| | Extension | −1.18 | 59 | 0.242 | −3.1 | −0.153 | −0.406 | 0.103 |
| Flexion | Extension | 3.4 | 59 | 0.001 | 9.28 | 0.44 | 0.173 | 0.703 |

Note. $H_a$ $\mu_{Measure\ 1\ -\ Measure\ 2} \neq 0$.

**Table 3 | Differences between real and estimated hand rotation values. One sample T-test**

| Hand | Movement | t | df | p | Mean diff. | d | Lower 95% CI | Upper 95% CI |
|---|---|---|---|---|---|---|---|---|
| Dominant | Adduction | −1.97 | 58 | 0.054 | −4.94 | −0.256 | −0.5146 | 0.00414 |
| | Abduction | 4.33 | 58 | <0.001 | 9.9 | 0.563 | 0.2863 | 0.8361 |
| | Flexion | 5.25 | 58 | <0.001 | 24.15 | 0.683 | 0.3969 | 0.96427 |
| | Extension | 2.77 | 58 | 0.008 | 11.41 | 0.36 | 0.0955 | 0.62237 |
| Nondom. | Adduction | −1.08 | 58 | 0.286 | −2.47 | −0.14 | −0.396 | 0.11683 |
| | Abduction | 5.44 | 58 | <0.001 | 12.1 | 0.708 | 0.4202 | 0.99168 |
| | Flexion | 4.5 | 58 | <0.001 | 22.52 | 0.586 | 0.3073 | 0.86024 |
| | Extension | 4.01 | 58 | <0.001 | 16.65 | 0.522 | 0.2481 | 0.79252 |

*Note.* $H_a$ $\mu \neq 0$.

**Table 4 | Differences between the absolute values of the actual and imagined wrist rotation amplitude in Experiment 2. One sample t-test**

| Hand | Movement | t | df | p | Mean diff. | d | Lower 95% CI | Upper 95% CI |
|---|---|---|---|---|---|---|---|---|
| Dominant | Adduction | 5.93 | 24 | <0.001 | 7.24 | 1.19 | 0.664 | 1.69 |
| | Abduction | 6.8 | 24 | <0.001 | 12.72 | 1.36 | 0.805 | 1.9 |
| Nondominant | Adduction | 6.64 | 24 | <0.001 | 9.56 | 1.33 | 0.779 | 1.86 |
| | Abduction | 6.7 | 24 | <0.001 | 15.72 | 1.34 | 0.789 | 1.88 |

*Note.* $H_a$ $\mu \neq 0$.

**Table 5 | Differences between real and imagined hand rotation values in Experiment 2. One sample t-test**

| Hand | Movement | t | df | p | Mean diff. | d | Lower 95% CI | Upper 95% CI |
|---|---|---|---|---|---|---|---|---|
| Dominant | Adduction | 0.957 | 24 | 0.348 | 1.8 | 0.191 | −0.2063 | 0.585 |
| | Abduction | 3.846 | 24 | <0.001 | 9.68 | 0.769 | 0.3154 | 1.211 |
| Nondominant | Adduction | 2.317 | 24 | 0.029 | 5.08 | 0.463 | 0.046 | 0.872 |
| | Abduction | 4.753 | 24 | <0.001 | 13.56 | 0.951 | 0.4693 | 1.418 |

*Note.* $H_a$ $\mu \neq 0$.

involved in the action. We suggest that this asymmetry in the consequences of planning an action based on accurate versus underestimated representation of one's range of motion might prevent joint limit violations and, in result, shapes humans' models of their body biomechanical limits available to cognition. One prediction of this hypothesis is that underestimation of movement range representation results from uncertainty in movement control. More generally, this idea is also supported by data from motor planning studies showing that human hand movements are undershot when executed under uncertainty which suggest that they are strategically under-planned unless execution conditions are predictable[17–19]. Although this prediction remains to be tested more directly, it is corroborated by fact that underestimations were always larger for the non-dominant hand, as it is usually less involved in precise movements. Therefore, precision of representations and precision of movements seem to be related.

A possible alternative explanation to our results could be that biased movement representations emerge from distorted neural representations in somatosensory cortex as reported earlier for hand position, shape and size[7]. It is conceivable that misrepresented metrics of the hand affect other, related sensorimotor domains, such as representations of joint ranges. These internal representations seem critical for predictive modelling of actions and efficient action monitoring that compares predicted and actual action outcomes[5,20,21]. As motor planning, motor imagery, action monitoring and internal models such as efference copy or body schema share their neural substrates, particularly in parietal cortex[5,6,20–22], it is probable that they all rely on such biased representations resulting in a distorted image of body motion range. Whether this is the case, remains to be tested.

## Limitations

While central tendencies clearly indicate underestimation bias, it is important to note that there was variability across movements and participants. Unlike the other three movements, we did not find evidence that the range of wrist adduction was underestimated across all conditions. We observed a significant effect of underestimating adduction only for the non-dominant hand in the motor imagery task (Experiment 2) although a trend was present also in Experiment 1. It could be that we failed to detect a consistent underestimation of the adduction range of movement simply because it is naturally the smallest of the four movements that we tested, as evident from that biases are bigger for larger movements (see Supplementary Table S1).

There was also some potentially interesting variability between the participants. Although, at a group level, there was a tendency to underestimate movement range, some subjects tended to overestimate. In our view, this between-subjects variability may be driven, at least in part, by individual differences in reliance on visual vs. proprioceptive information in the construction of body movement representation, as known from other domains that require binding between different sensory input sources, e.g. in visuomotor realignment[23,24]. Moreover, while our joints, including our wrists, are capable of a wide range of motion, the demands of daily life activities generally require only a fraction of that capacity. This limited use of the full movement range may lead people relying primarily on visual information to a more conservative estimate of what can be done. In contrast, individuals relying primarily on proprioceptive feedback may get a better sense of the full range of motion by accessing additional information

such as the feeling of muscle stretch or joint strain. In our framework, another source of variability may lie in individuals' different trade-offs between the cost of movement undershoot and the risk of movement overshoot. In turn, these differences may relate to different factors such as individuals' different motor control skills and experience and variations in proprioceptive ability. For instance, for the same reason that people underestimate less the range of movement of their dominant hand, individuals with better motor control skills and experience may be more precise and therefore less prone to underestimate their limb's range of movement.

Future studies should determine, to what degree the estimation of movement range rely on vision and proprioception but also internal models such as body schema or efference copy as all these input sources seem critical for action planning and monitoring[5,6,20,22]. In sum, further research is needed to better understand the complex interplay of cognitive and physiological factors that, together, contribute to these observed biased movement representations, and how these representations differ between movements and individuals.

## Conclusions

We report that representations of body movements seem to underestimate one's real biomechanical limits. We suggest that this bias optimizes movement efficiency while ensuring safety despite the inherently stochastic nature of movement planning and execution.

## Data availability

Data is available for download from osf.io/gzrj7.

## Code availability

Code is available for download from osf.io/gzrj7.

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

## Acknowledgements

This research was supported by an *Action de Recherche Concertée* from the Université catholique de Louvain to GV, and Bial Foundation grant 260/22 to AP. The funders had no role in study design, data collection and analysis, decision to publish or preparation of the manuscript.

## Author contributions

A.P., A.V. and G.V. conceived the study and designed the experiments. All authors designed and constructed equipment. G.A. and A.V. collected the data. All authors analysed the data. A.P., A.V. and G.V. wrote the paper. All authors approved the final version of the paper.

## Funding

## Competing interests

The authors declare no competing interests.
