## [Transparent Peer Review file · Communications Psychology]

Humans underestimate the movement range of their own hands

Corresponding Author: Dr Artur Pilacinski

Version 0:

Decision Letter:

Dear Dr Pilacinski,

Thank you for your patience during the peer-review process. Your manuscript titled "Systematic underestimation of hand movement range in humans" has now been seen by 2 reviewers, and I include their comments at the end of this message. They find your work of interest but raised some important points. We are interested in the possibility of publishing your study in Communications Psychology, but would like to consider your responses to these concerns and assess a revised manuscript before we make a final decision on publication.

We therefore invite you to revise and resubmit your manuscript, along with a point-by-point response to the reviewers. Please highlight all changes in the manuscript text file.

Editorially, we consider it important that the revised manuscript provides an introduction that better conceptually or theoretically motivates the research question/hypotheses. If hypotheses were specified a priori, please include them at the end of the introduction. Otherwise, please state the research question. Likewise, the revised manuscript should provide an improved discussion that better situated the findings within the existing literature. Alternative explanations for the findings should be clarified with additional analyses or experimentation.

Please ensure you follow our statistical guidelines when reporting statistics (<https://www.nature.com/commspsychol/submit/submission-guidelines#statistical-guidelines>). Please note in particular our requirements for the reporting and interpretation of null-results. Non-significant findings derived from null-hypotheses significance tests should be reported in full, but may not be interpreted. Where you interpret null results, this interpretation must be based on Bayes Factors or equivalence tests.

I am attaching an Editorial Requests Table that details critical reporting requirements for the revised manuscript. Please attend to each item and ensure your manuscript is fully compliant. We are requesting that your manuscript aligns with these requirements as this facilitates the evaluation of your manuscript, reducing delays in re-review and potential future acceptance. If your revised manuscript is not aligned with these requests on major issues, such as those concerning statistics, it may be returned to you for further revisions without re-review. Additional information can be found in our style and formatting guide <https://www.nature.com/documents/commspsychol-style-formatting-guide-accept.pdf> Communications Psychology formatting guide.

Please use the following link to submit your

- revised manuscript,
- point-by-point response to the referees' comments,
- cover letter (as a separate document),
- the Editorial Policy Checklist (see below),
- the Reporting Summary (see below), and
- the completed Editorial Request Table (attached):

Link Redacted

Best regards,

Jennifer Bellingtier

Jennifer Bellingtier, PhD
Senior Editor
Communications Psychology

REVIEWER EXPERTISE:

Reviewer #1 motor movement

Reviewer #2 motor imagery

REVIEWER REPORTS:

Reviewer #1 (Remarks to the Author):

Undershooting may be well known for saccadic eye movements, but not for arm movements. Ataxia in impaired individuals has been characterized by both undershooting and overshooting, and overall, across humans and experiments, errors can be found all over the place, including left and right. Moreover, studies on reachability generally report that humans overestimate how far they can reach. While this may not be systematic, there is plenty of evidence of biases in the human sensorimotor system, and thus plenty of evidence that internal representations are not necessarily veridical, even in healthy, young adults. The major claim of the paper, 'Systematic undershooting', thus lacks overwhelming support in the literature. Although the authors collected a nice amount of data, they do not support the idea of 'Systematic undershooting'. Indeed, undershooting is found in 6 out of 8 conditions, which is an interesting result but does not qualify as a systematic finding. Also the data are plotted with histograms which do not show individual data. The expectation from the title is to see that underestimation is found in all participants.

While a great number of participants was tested, only three measures per condition were taken in the first experiment: the 24 measures per participant amount to a rather limited set of data. Only two measures per condition were taken in the second experiment. Could the authors justify such choices?

Overall, I failed to see how this manuscript could represent a substantial advance in understanding which may influence thinking in the field. Part of this may simply be due to formatting: for instance, in the abstract and introduction, the authors did not present the question they asked with their experiment(s). Also the literature review is extremely brief and there was no specific hypothesis. I believe reporting these missing elements would strengthen the manuscript.

The authors did not explain why they tested different conditions, and what were the hypotheses.

The authors should explain in the main text how estimation was assessed in the first experiment. The authors should also clarify the logic of running experiment 2 on motor imagery. I was interested in the real and imagined movements, but it was difficult to connect the two experiments. It was also difficult to understand the link between motor imagery and motor planning: the authors should adjust their manuscript to facilitate such understanding.

At last, the authors should discuss the issue of lateralization that they chose to tackle.

They may also speculate on which signal (visual, proprioceptive, and/or efference copy: see Wolpert et al. 1995 Science) may have contributed most to the bias observed in most conditions.

Reviewer #2 (Remarks to the Author):

The authors examine the relationship between the actual movement range and imagined movement range in two experiments. In both, they find that participants consistently underestimate all movement types examined (abduction, extension and flexion) apart from adduction. Furthermore, they find a consistent effect such that underestimations were

larger for the non-dominant hand when compared to the dominant hand. The authors interpret the findings as resulting from balancing movement efficiency and safety. The results are clear and compelling. However, I have some concerns regarding the interpretation of the results.

First, the authors interpretation isn't completely consistent with their findings. If these findings are about a balance between movement efficiency and safety, then one would expect it to be observed in all four types of movement. The fact that this was not observed for adduction either suggests an alternative explanation, or that there is some bias specific to adduction that is masking the underestimation effect. Furthermore, it's not clear why conscious biases about biomechanical constraints would necessarily relate to movement efficiency/safety. This would be strengthened if there were results that showed that these biases actually related to movement kinematic in some way (e.g., perhaps there is a relationship between actual movement efficiency and these results, such that on some tasks, extension [the task w/the most underestimation] would differ from adduction [the task w/the least underestimation]).

Second, the results in relation to "traditional views" are a little overstated. The authors state that their findings challenge "the traditional views on motor imagery...as veridically representing biomechanical limits of what out (sic) limbs can do." I am not aware of models that claim that motor imagery is *perfectly* veridical, and it is not clear that "traditional views" on motor imagery would preclude such a finding.

EDITORIAL POLICIES

We ask that you ensure your manuscript complies with our editorial policies and reporting requirements.

To that end, we require revised manuscripts to be accompanied by two completed items: a reporting summary that collects information on study design and procedure, and an editorial policy checklist that verifies compliance with all required editorial policies.

- <https://www.nature.com/documents/nr-reporting-summary.zip>>Nature Research Reporting Summary
- <https://www.nature.com/documents/nr-editorial-policy-checklist.pdf>>Editorial Policy Checklist

All points on the policy checklist must be addressed. Your revised manuscript can only be sent back to the referees if these checklists are completed and uploaded with the revision.

Notes: If you have submitted a Stage 1 Registered Report, Review, Primer, Comment, or Perspective you do not need to submit these forms. If you have already submitted these forms, you may disregard this request.

** Visit Nature Research's author and referees' website at <http://www.nature.com/authors>>www.nature.com/authors for information about policies, services and author benefits**

If you experience problems in linking your ORCID, please contact the <http://platformsupport.nature.com/>>Platform Support Helpdesk.

Version 1:

Decision Letter:

Dear Dr Pilacinski,

Your manuscript titled "Underestimation of own hand movement range in humans" has now been seen by our reviewers, whose comments appear below. In light of their advice I am delighted to say that we are happy, in principle, to publish a suitably revised version in Communications Psychology.

We therefore invite you to revise your paper one last time to address the remaining concerns of our reviewers and a list of editorial requests. At the same time we ask that you edit your manuscript to comply with our format requirements and to maximise the accessibility and therefore the impact of your work.

EDITORIAL REQUESTS:

SUBMISSION INFORMATION:

OPEN ACCESS:

*** TRANSPARENT PEER REVIEW:** Communications Psychology uses a transparent peer review system. On author request, confidential information and data can be removed from the published reviewer reports and rebuttal letters prior to publication. If you are concerned about the release of confidential data, please let us know specifically what information you would like to have removed. Please note that we cannot incorporate redactions for any other reasons.

*** CODE AVAILABILITY:** All Communications Psychology manuscripts must include a section titled "Code Availability" at the end of the methods section. We require that the custom analysis code supporting your conclusions is made available in a publicly accessible repository at this stage; please choose a repository that generates a digital object identifier (DOI) for the code; the link to the repository and the DOI must be included in the Code Availability statement. Publication as Supplementary Information will not suffice.

*** DATA AVAILABILITY:**

Link Redacted

Best regards,

Jennifer Bellingtier

Jennifer Bellingtier, PhD
Senior Editor
Communications Psychology

REVIEWERS' EXPERTISE:

Reviewer #2 motor imagery

REVIEWERS' COMMENTS:

Reviewer #2 (Remarks to the Author):
no further comments

Dear Editor,

Dear Reviewers,

First off, we would like to thank both Reviewers for their constructive comments that in our opinion helped substantially improving our manuscript on several levels. Specifically, we want to thank Reviewer 1 for their feedback on theoretical framing of our work (which we agree was too rudimental) and Reviewer 2 for their feedback on our interpretation of the results (which we find valid and, likewise, tried to reflect it in the revision). Please find below in blue text our responses to individual *comments from the Reviewers*.

REVIEWER REPORTS:

Reviewer #1 (Remarks to the Author):

Undershooting may be well known for saccadic eye movements, but not for arm movements. Ataxia in impaired individuals has been characterized by both undershooting and overshooting, and overall, across humans and experiments, errors can be found all over the place, including left and right.

Moreover, studies on reachability generally report that humans overestimate how far they can reach. While this may not be systematic, there is plenty of evidence of biases in the human sensorimotor system, and thus plenty of evidence that internal representations are not necessarily veridical, even in healthy, young adults. The major claim of the paper, 'Systematic undershooting', thus lacks overwhelming support in the literature. Although the authors collected a nice amount of data, they do not support the idea of 'Systematic undershooting'. Indeed, undershooting is found in 6 out of 8 conditions, which is an interesting result but does not qualify as a systematic finding. Also the data are plotted with histograms which do not show individual data. The expectation from the title is to see that underestimation is found in all participants.

We agree with this comment. Accordingly, we (1) modified the title and wording throughout the manuscript, to avoid confusion. (2) updated the main results figure to a box-and-whiskers plot to better reflect this variability, and (3) added some discussion of the possible variability across tasks and individuals as by itself it may be an interesting finding.

While a great number of participants was tested, only three measures per condition were taken in the first experiment: the 24 measures per participant amount to a rather limited set of data. Only two measures per condition were taken in the second experiment. Could the authors justify such choices?

This is a good question as we debated this before conducting the experiment. We decided for a low number of measures as: a) we found that our subjects were consistent in their estimations (average size of difference between repetitions was less than 3 deg.) and b) we wanted to avoid demand characteristics or other cognitive factors influence subjects' behaviour as often encountered in repetitive tasks. We also tried to at least partially compensate for such factors through explicitly asking our subjects if they used any cognitive strategy during the experiment, but nobody reported such a thing.

Overall, I failed to see how this manuscript could represent a substantial advance in understanding which may influence thinking in the field. Part of this may simply be due to formatting: for instance, in the abstract and introduction, the authors did not present the question they asked with their experiment(s). Also the literature review is extremely brief and there was no specific hypothesis. I believe reporting these missing elements would strengthen the manuscript.

Thank you for the comment. We agree that the previous framing could be a bit confusing, and we now have reworked our introduction and discussion to make the two main contributions of our study clearer:

- (1) We contribute the first data set relevant to the question whether the representation of our joints' range of movement is veridical or not? How precise is it? These data are important for any future theory of motor planning or imagery. Specifically, and according to your comments below, we rephrased how motor planning and imagery link to the aim of our study in that they both rely on conscious representations of movements and as such seem considered to veridically represent movement limits [ref. 1,2].
- (2) We report that representation of our own range of movement is often underestimated. Distortions and biases have a long history of providing a window into the working of the mind/brain and more recent studies show that our body image and static properties are distorted (ref. 7,9), shedding a new light on human conscious representations of their body (ref. 8). By reporting this new bias in movement range representation, we hope to contribute new insight about the constraints within which action planning and movement representations operate and the trade-offs that may e.g. underlie its efficiency.

The authors did not explain why they tested different conditions, and what were the hypotheses.

Done. We agree this was missing from the previous version of the text which we aimed to keep as succinct as possible and might have resulted in the lack of clarity. We now unpacked on our hypotheses in the introduction of our manuscript, but we also rephrased a few other passages to better explain our reasoning.

The authors should explain in the main text how estimation was assessed in the first experiment.

Done.

The authors should also clarify the logic of running experiment 2 on motor imagery. I was interested in the real and imagined movements, but it was difficult to connect the two experiments. It was also difficult to understand the link between motor imagery and motor planning: the authors should adjust their manuscript to facilitate such understanding.

Done. We have clarified that Experiment 2 "allowed exploring the properties of hand movement range representations used when participants are engaged in explicit motor imagery, which is often assumed to reflect motor plans without over movements [1,2]."

On a side note, since you mentioned being interested in the comparison between real and imagined movements, we now also included a table containing descriptives of all our measures of both imagined and real wrist movements. This may be informative for more readers.

At last, the authors should discuss the issue of lateralization that they chose to tackle.

Thank you, we now included this in the paper. In brief, we expected that sensorimotor precision and/or practice related to handedness might affect the precision of estimations in our tasks. For this reason, we also asked a screening question regarding whether participants perform any manual activities that could affect their body movement awareness (such as stretching exercises). We now have added information about this in the manuscript.

They may also speculate on which signal (visual, proprioceptive, and/or efference copy: see Wolpert et al. 1995 Science) may have contributed most to the bias observed in most conditions.

This is a great point, which we now address in the Discussion section as it is also a potential source of interindividual differences. For example, it is known that different people differently rely on visual vs. proprioceptive cues e.g. in sensorimotor realignment (e.g. Block and Bastian, 2011; Block and Liyu, 2023), and we likewise think that the cognitive representations of movement range might differentially reflect this reliance on signals from different sources (including efference copy and/or body schema).

Reviewer #2 (Remarks to the Author):

The authors examine the relationship between the actual movement range and imagined movement range in two experiments. In both, they find that participants consistently underestimate all movement types examined (abduction, extension and flexion) apart from adduction. Furthermore, they find a consistent effect such that underestimations were larger for the non-dominant hand when compared to the dominant hand. The authors interpret the findings as resulting from balancing movement efficiency and safety. The results are clear and compelling. However, I have some concerns regarding the interpretation of the results.

Thank you for the feedback. We agree that we should have more carefully considered our interpretations and now have revised the wording throughout the manuscript and provided more information to improve clarity where we found necessary. Please find also our more specific replies below.

First, the authors interpretation isn't completely consistent with their findings. If these findings are about a balance between movement efficiency and safety, then one would expect it to be observed in all four types of movement. The fact that this was not observed for adduction either suggests an alternative explanation, or that there is some bias specific to adduction that is masking the underestimation effect.

We agree that this is a puzzling effect. To our understanding, the actually smaller size of adduction movement made it more difficult for the subjects to underestimate (as evident when comparing to larger amplitude movements such as flexion and extension or even abduction). We now added this point in the discussion as we find it a valid problem for future research on the actual involvement of different signal sources in performing these estimates.

Furthermore, it's not clear why conscious biases about biomechanical constraints would necessarily relate to movement efficiency/safety. This would be strengthened if there were results that showed that these biases actually related to movement kinematic in some way (e.g., perhaps there is a relationship between actual movement efficiency and these results, such that on some tasks, extension [the task w/the most underestimation] would differ from adduction [the task w/the least underestimation]).

This is a good point. In our view, our reasoning is supported by the studies showing undershoot bias is higher when planning movements under uncertainty, which indicates that the natural tendency is to undershoot a reach rather than overshoot if a movement is uncertain (References 10-12). As for our own data, we show that underestimation is substantially lower for the dominant hand, hinting us that experience with using a hand makes its movement more predictable and thus its conscious representation more veridical (though still biased). We now make this reasoning clearer in the Discussion.

Second, the results in relation to "traditional views" are a little overstated. The authors state that their findings challenge "the traditional views on motor imagery...as veridically representing biomechanical limits of what our (sic) limbs can do." I am not aware of models that claim that motor imagery is *perfectly* veridical, and it is not clear that "traditional views" on motor imagery would preclude such a finding.

We agree and should have phrased our point of view differently to not imply a consensus. We now changed the wording to also reflect the nuanced point of view in our manuscript and avoid strong statements about the putative "traditional views". We also believe that the problem of how accurate the internal models and conscious representations of body biomechanics are in general, a fascinating research topic by itself. For example, our paper shows that conscious representations of movement are inaccurate. How do these inaccuracies in conscious representations relate to non-conscious models (e.g. employed by motor synergies) is still to be determined.